# Medical advocacy in the face of Australian immigration practices: A study of medical professionals defending the health rights of detained refugees and asylum seekers

**Rohanna Stoddart**[1]*, **Paul Simpson**[2], **Bridget Haire**[2]

**1** Faculty of Medicine, University of New South Wales, Sydney, New South Wales, Australia, **2** The Kirby Institute, University of New South Wales, Sydney, New South Wales, Australia

\* rohanna.stoddart@outlook.com, z5120687@ad.unsw.edu.au

**Data Availability Statement:** Data within the manuscript and its supporting information file is not the minimal underlying data set. There are

## Abstract

While medical advocacy is mandated as a core professional commitment in a growing number of ethical codes and medical training programs, medical advocacy and social justice engagement are regularly subordinated to traditional clinical responsibilities. This study aims to provide insight into factors that motivate clinician engagement and perseverance with medical advocacy, so as to inform attempts by policymakers, leaders and educators to promote advocacy practices in medicine. Furthermore, this study aims to provide an analysis of the role of medical advocates in systems where patients' rights are perceived to be infringed and consider how we might best support and protect these medical advocates as a profession, by exploring the experiences and perspectives of Australian clinicians defending the health of detained asylum seekers. In this qualitative study thirty-two medical and health professionals advocating on asylum seeker health in immigration detention were interviewed. Transcripts were coded both inductively and deductively from interview question domains and thematically analysed. Findings suggested that respondents' motivations for advocacy stemmed from deeply intertwined professional and personal ethics. Overall, advocacy responses originated from the union of three integral stimuli: personal ethics, proximity and readiness. We conclude that each of these three integral factors must be addressed in any attempt to foster advocacy within the medical profession. In light of current global trends of increasingly protectionist immigration practices, promoting effective physician advocacy may become essential in ensuring patients' universal right to health.

## Introduction

Medical advocacy is increasingly mandated as a core component of medical professionalism and ethical practice, featuring in a number of codes of conduct [1]. Today's physicians are being called upon to engage with public health discourses and promote social justice, to be responsive to communities, to consider social determinants of health, and to provide holistic

ethical restrictions, endorsed by the UNSW Sydney Human Research Ethics Committee, on making the data publicly available. This is due to participant confidentiality issues, as some of the data that is not within the manuscript has the potential to (re) identify participants. Participant consent was not sought for the minimal underlying data set to be publicly available. Non-author contact regarding any future inquiries is the University of New South Wales ethics secretariat who can be contacted on: +61 2 9385 6222, + 61 2 9385 7257 or + 61 2 9385 7007 (quote reference number HC180778). Alternatively the UNSW ethics office can be emailed at: humanethics@unsw.edu.au Data will be stored for 5 years from the date of publication, after which electronic files will be deleted, and hard copies of informed consent forms will be shredded.

**Funding:** The author(s) received no specific funding for this work.

**Competing interests:** The authors have declared that no competing interests exist.

care to their patients [2, 3]. Both the medical profession and the wider public have emphasised the importance of training physician advocates, and there is a growing focus on health advocacy in medical training programs [4].

What is meant by advocacy, and what is expected from clinicians in terms of promoting social justice, however, remains unclear [1, 5]. The majority of doctors do not engage in medical advocacy, even though they identify it as a core professional commitment in surveys [6, 7]. Thus, while clinicians feel that they *should* view advocacy as central to their role, in practice they do not: medical advocacy is subordinated to traditional clinical responsibilities [8]. If it is desirable to promote competent physician advocacy, then analyses of medical professionals' reasons behind advocacy involvement are imperative for informing medical ethics teaching. This is an area of notable paucity in the literature, and one that will be addressed in this study.

For the purposes of this study, the definition of medical advocacy as proposed by Earnest et al. [1] will be used, where advocacy refers to:

'Actions by a [clinician] designed to promote those social, economic, educational and political changes that ameliorate the suffering and threats to human health and wellbeing that he or she identifies through his or her professional work and expertise' [1].

This includes addressing social determinants of health to promote social justice and improve the health-outcomes of communities and populations. Notably, this is held to be distinct from micro-level advocacy, delineated by Dobson et al. [5] as advancing the health of individual patients. While both fall under the rubric of health advocacy, physician advocacy for individual patients is placed firmly within, and widely accepted as, professional duty; whereas medical advocacy and its scope are less clearly defined in bioethical discourses [2].

## Background

### Australian immigration detention and the health of detained asylum seekers

Under the Australian Migration Act 1958, asylum seekers can be detained indefinitely in prison-like facilities without judicial oversight [9, 10]. This includes mandatory detention in offshore processing centres, a practice which has been widely condemned as violating both human rights and international law [11].

In the setting of Australian immigration processing, indefinite detention in poor conditions has serious and long-standing impacts on physical and mental health [12–17]. In particular, the detention environment is shown to be strongly associated with patterns of deteriorating psychological health [10], and a growing body of clinical evidence supports the view that immigration detention itself is implicated in the aetiopathogenesis of psychiatric disorders [12, 15–17]. Australian immigration policy is therefore as much a public health challenge as a political issue.

### Healthcare delivery in offshore immigration detention: A systemic failing

Within offshore processing centres, health services are provided through independent contractors of the Australian government. Since 2007, this has been the International Health and Medical Services company (IHMS), which is contractually obliged to: '. . .provide a level of healthcare to people in immigration detention consistent with that available to the wider Australian community. . .' [18]

However, repeated failures to meet objective standards of patient care have been widely reported in clinical accounts and commissions of inquiry [19, 20]. These relate to substandard detention conditions: including reports of riots, deaths from violence, poor living-conditions

and wide-spread physical and sexual abuses (some involving children), and to failures in the delivery of standard medical care: including reports of insufficient resources, poor facilities, rapid processing of detainees at the expense of standard clinical assessment, deaths from medical neglect and epidemic levels of self-harm and parasuicides [21–27]. Further, government policies of indefinite detention have actively opposed attempts to meet standards of care, and legislative measures have been taken to reduce oversight and tighten secrecy around healthcare services in offshore processing centres [11, 24, 28–30].

The Australian Department of Immigration and Border Protection (DIBP) has also been reported to regularly challenge clinical recommendations, particularly those advising emergency medical transfer from offshore detention. Clinicians working in immigration detention centres reported their medical recommendations commonly being ignored, delayed or dismissed by IHMS and the DIBP, with adverse consequences for patient health outcomes [11, 31].

## Ethical and moral challenges of immigration detention

The involvement of medical professionals in Australian immigration detention raises complex clinical and ethical issues, not least because clinician independence is likely to be compromised by contractual obligations to IHMS and the DIBP [29, 32–35]. Dual loyalty describes circumstances in which the best interests of patients compete with other obligations, such as those to a third party or employer [36]. Such conflicts are regularly faced by clinicians working in Australian immigration detention centres (IDCs), where the expectations of IHMS and the immigration department do not always align with patient interests or with reasonable standards of patient care [29, 32–35].

Bioethics literature is increasingly engaging with ethical issues faced by clinicians working within IDCs, calling for clear, directional guidelines that specifically address the issues at hand [29, 33]. Some authors argue that the presence of healthcare services in immigration facilities provides a veneer of respectability that facilitates systemic abuses [33], and that medical professionals are thusly complicit in the wrongdoings of the DIBP [28, 37]. Others take this further to insist that a boycott is ethically necessary [30, 38]. The literature in this area overwhelmingly originates from a subset of clinicians engaged in medical advocacy and whistleblowing who draw on their personal experiences of immigration detention. To date, no study has evaluated the perceptions and ethical motivations of such medical advocates, and an opportunity exists here to assess the position and reflexivity of these actors.

## Medical ethics

In order to understand medical advocates' responses to the issues at hand, it is useful to examine medical ethics and their origins. Any such discussion customarily begins with the principles outlined in *The Hippocratic Oath* of the 4th century BC, which remains a source of professional inspiration to this day (Table 1). Most importantly are the embedded values of beneficence and non-maleficence, famously represented in the tenet of, 'first, do no harm.' These values, along with justice and patient autonomy, are represented in other codified forms of medical ethics, including principlism [39] and virtue ethics [40]. The Nuremburg trials were a further galvanising force of normativity in bioethics, providing the forum for the development of normative ethical standards. The resultant *Nuremburg Code* builds on the idea of a Hippocratic duty to inform clinical research ethics and forms the basis for all modern ethical codes and international human rights frameworks related to health, medicine and medical research [41].

**Table 1. Normative tools that can apply to medical ethics in Australian offshore detention.**

| Normative tool | Year [1] | Issuing body / Jurisdiction | Relevant recommendations | Are doctors obligated to uphold human rights?* | Normative power* / Is it enforceable?* | Does it address medical advocacy?* | Does it address immigration detention?* |
|---|---|---|---|---|---|---|---|
| **Hippocratic Oath** | 400$_{BC}$ | Hippocrates of Kos / Not applicable | I will come for the benefit of the sick, remaining free of all intentional injustice [42]. | Not applicable | Low / No | Not in classical version | No |
| **Declaration of Geneva** | 1948 (2017) | WMA: World Medical Association / International Adopted by AMA | The health and wellbeing of my patient will be my first consideration. I will not use my medical knowledge to violate human rights and civil liberties, [43]. | Yes[2] [44] | High / No | No | No |
| **Declaration of Helsinki** | 1964 (2013) | WMA: World Medical Association / International | Ethical guidelines specific to medical research, that still holds general principles of the "duty of the physician": 3. to act in the patients' best interests, and 4. to promote and safeguard the health, wellbeing and rights of patients, [45]. | Yes, over any other national or international law or requirement. | High / No | Yes Refer to: 4 | No |
| **International Code of Medical Ethics** | 1949 (2006) | WMA: World Medical Association / International | A physician shall act in the patient's best interest…and, … provide competent medical service in full professional and moral independence, with compassion and respect for human dignity [46]. | Not mentioned | High / No | No | No |
| **United Nations Principles of Medical Ethics** | 1982 | Office of the High Commissioner of Human Rights / International | *Principle 2.* Physicians cannot, "engage, actively or passively, in acts which constitute participation in, complicity in, incitement to or attempts to commit torture or other cruel, inhuman or degrading treatment or punishment" [47]. | Yes | High / Yes, under human rights treaties[3] [48] | Not explicitly mentioned. | Yes Refer to: Principle 1 |
| **Good Medical Practice: A Code of Conduct for Doctors in Australia** | 2014 | MBA: Medical Board of Australia / Australia | 1.4 Doctors have a duty to make the care of patients their first concern. 1.4 Doctors have a responsibility to protect and promote the health of individuals and the community [43]. | Doctors are encouraged to respect human rights, but legislative law takes precedence over the code. | High / Yes, misconduct can result in reprimand or deregistration by the Medical Board of Australia[4] [49]. | Yes Refer to: 5.3 on health advocacy. | No |

*(Continued)*

**Table 1.** (Continued)

| Normative tool | Year [1] | Issuing body | Relevant recommendations | Are doctors obligated to uphold human rights?* | Normative power* | Does it address medical advocacy?* | Does it address immigration detention?* |
|---|---|---|---|---|---|---|---|
| | | Jurisdiction | | | Is it enforceable?* | | |
| **AMA Code of Ethics** | 2004 (2016) | AMA: Australian Medical Association<br><br>Australia | 2.1.1 Consider <u>first</u> the wellbeing of the patient. 4.2.1 Uphold professional autonomy and clinical independence and <u>advocate</u>. . . [for this] in the treatment of patients <u>without undue influence</u> by individuals, governments or third parties. 4.2.2 Refrain from entering into any <u>contract</u>. . . which you consider may <u>conflict with</u> your <u>professional autonomy</u>, <u>clinical independence</u> or your <u>primary obligation</u> to the <u>patient.</u> 4.2.3 Recognise your <u>right to refuse</u> to carry out services which you consider to be <u>professionally unethical</u>, against your moral convictions. . . or which you consider are not in the <u>best interests of the patient.</u> 4.2.5 Contemporary protections for <u>whistleblowers</u> should be supported by doctors [50]. | Yes[3] [48] 4.6.3 Do not countenance, condone or participate in the practice of torture or other forms of cruel, inhuman or degrading procedures. | High<br><br>Yes, misconduct can result in reprimand or deregistration by the Medical Board of Australia[4] [49]. | Yes Refer to: 4.2.1 re. advocating for clinical independence 4.2.3 re. boycotting, and both 4.2.5, as well as AMA position policy [51] Article 16 re. whistleblowing | Yes Refer to: AMA's Policy on Health Care of Asylum Seekers and Refugees [51]. |
| **Dual Loyalty and Human Rights in Health Professional Practice** | 2002 | Physicians for Human Rights<br><br>International | Health professionals: 3.3. . . .must place the protection of the patient's human rights and wellbeing first whenever there exists a conflict between the patient's human rights and the state's interests. . . 3.9. . . .should take all possible steps to resist state demands to participation in a violation of the human rights of patients 5.7.2. Advocacy to change laws and regulations that prevent or impede health professionals from meeting their human rights obligations to patients. [52] | Yes Refer to: Guidelines 1, 3, 5, 7, 9, 10, 11, 12 and 13. | Medium<br><br>No | Yes. Refer to: 5.7 *Collective action by the Professions* | Yes. Refer to: 4 (B) *Guidelines on Health Care for Refugees and Immigrants* |

(*Continued*)

**Table 1.** (Continued)

| Normative tool | Year [1] | Issuing body | Relevant recommendations | Are doctors obligated to uphold human rights?* | Normative power* | Does it address medical advocacy?* | Does it address immigration detention?* |
| --- | --- | --- | --- | --- | --- | --- | --- |
| | | Jurisdiction | | | Is it enforceable?* | | |
| **Universal Declaration on Bioethics and Human Rights** | 2005 | UNESCO: United Nations Educational, Scientific and Cultural Organisation | 3.1 Human dignity, human rights and fundamental freedoms are to be fully respected.18.2 Persons and professionals concerned and society as a whole should be engaged in dialogue on a regular basis.18.3 Opportunities for informed pluralistic public debate, seeking the expression of all relevant opinions, should be promoted [53]. | Yes Cites all International Human Rights instruments of the United Nations. | High | Yes Refer to: articles 18 and 22.1. | Implicit in articles 10, 11 and 28. |
| | | International | | | No | | |

[1] When there are multiple revisions by an issuing body or assembly then the original date and the date of the most recent revision, this latter bracketed, are noted.

* The assessment columns are included to evaluate the strength, applicability and usefulness of these normative tools for clinicians attempting to navigate Australian immigration detention.

[2] The right to health has been defined as "the right of everyone to the enjoyment of the highest attainable standard of physical and mental health" in the International Covenant on Civil and Political Rights at the 1966 United Nations General Assembly.

[3] The Australian Government ratified the United Nations' Optional Protocol to the Convention against Torture (OPCAT) on 15 December 2017.

[4] Doctors engaging in professional misconduct are subject to investigation by a national board under the Health Practitioner Regulation National Law Act 2009.

## Normative tools of medical ethics and intersections of international human rights

Discussions of refugee and asylum seeker health are inseparably linked with international law, as the 'right to health' is codified in the Universal Declaration of Human Rights [54] and international covenants and treaties. These legal instruments are used to support and inform theories of medical ethics in public health literature, whilst more traditional codes focus on the 'physician's role'. In order to better understand the evidence available for clinicians attempting to navigate and reconcile ethical dilemmas of Australian immigration policy, Table 1 has been constructed to critically analyse normative tools of medical ethics and how they can be applied to immigration detention. Not intended to be an exhaustive list, the instruments presented in Table 1 were chosen for their normative power and relevance to Australian clinicians.

## Rationale

In 2015 the United Nations Refugee Agency announced that they were seeing the highest levels of displacement on record [55]. This growing global crisis has been paralleled by increasingly protectionist approaches to immigration, with detention of refugees and asylum seekers institutionalised in migration policies worldwide [10]. While Australia is notable in the extremity of its restrictive policies and anti-migrant rhetoric, recent expansion of mandatory detention in the United States [56], and the rise of national frontiers and temporary border checks within Europe's Schengen area [57], indicate the globalisation of these policies. Accordingly, research exploring the role of medical advocates in immigration detention is important and timely.

This study focuses on Australian offshore detention as an acute setting for human-rights abuses, and as a noted focus of past medical activism movements. By investigating and documenting ethical responses to a system that expects clinicians to deliver diminished standards of care and violate patients' rights to health, this study will make an important contribution to the literature on bioethics and medical professionalism. Furthermore, by examining motivating factors for participants' engagement with medical advocacy, this study will also provide insight into how medical schools can foster competent physician advocacy in their students. The need for analysis of medical advocate participants' understandings in their own words means that an in-depth qualitative approach has been necessary for assessment.

## Research aims

1. To document the personal views and experiences of Australian medical and health professionals advocating for the health of refugee and asylum seekers populations in mandatory immigration detention,

2. To explore how this response reflects contemporary medical ethics and understandings of core professional roles,

3. To provide insight into factors that motivate clinician engagement and perseverance with medical advocacy, and,

4. To examine the role of medical advocates in systems that diminish patients' health rights.

## Methodology

The conceptual framework for this study is grounded in empirical bioethics, which is an interdisciplinary approach that integrates empirical social science methods with ethical analysis to draw normative conclusions [58]. Data was collected through semi-structured interviews; a foundational and widely used method in qualitative research [59] which facilitates rich insights into subjective perspectives of experience and enables exploration and critical reflection on ethically complex topics [60]. Our analytical approach draws on reflexive thematic analysis of Braun and Clark, which recognises that qualitative analysis is an interpretive process informed by the research team's assumptions, values and commitments [60–62].

Our epistemological perspective is constructivist [63]. Study rigour is demonstrated by the selection of sampling, data collection and analysis methods that were best suited to answering our research questions [64].

Approval from UNSW Sydney's HREC was received for this study (HC180778) and included the use of both written and verbal consent. Data will be stored for 5 years from the date of publication, after which electronic files will be deleted, and hard copies of informed consent forms will be shredded. The study is presented in line with Consolidated criteria for reporting qualitative studies (COREQ) guidelines [65].

### Participant recruitment

Study participants were identified through professional networks and literature searches of journals and wider media. The first author is a medical student active in the movement for asylum seeker health rights, which facilitated recruitment as she had an established network within the community from which participants were drawn. Most study participants already knew the first author, and those who did not were informed about her role in the movement. Participants were purposively selected and invited to participate by email. Snowball sampling

was useful in this study, both in its efficacy for targeted recruitment and in uncovering the hidden social experiences of a niche population subgroup [66]. Data collection was from May 2019 to August 2019. Over these four months, 46 potential participants were directly contacted, and 34 initially responded. After follow-up, 32 participants were interviewed. Of the non-participants (n = 14); only one individual declined to be involved in the study, citing time-pressures. One potential respondent initially agreed but was lost to follow up, and the others (n = 12) did not respond to publicly available or participant provided emails and were otherwise uncontactable.

We selected participants who:

1. have participated in advocacy for asylum seekers, and

2. are currently medical or health professionals (doctors, nurses, mental health and allied health practitioners), and/or fulltime students in the listed professions.

It is important to note here that the study is focused mainly on medical doctors and medical training, to the exclusion of other health professions. We felt it was important to include the voices of health rights advocates from all health professions who had published influential papers or were otherwise leading advocates for asylum seeker policy reform. These participants spoke upon their personal experiences advocating for the health of refugee and asylum seekers populations in mandatory immigration detention, and their perspectives on medical advocacy generally. The vast majority of participants (28 of 32), however, were medical doctors or students, and this is reflected in the results, discussion and conclusion.

## Data collection

Data was collected through semi-structured, in-depth interviews, either face-to-face or by telephone (Table 2). All interviews were conducted by the first author, who is an undergraduate medical student, a cis-gender woman and an activist for asylum seeker health rights. She received training in qualitative interviewing from the third author, who is an experienced qualitative research and also a cis-gender woman. Pilot interviews were conducted with staff at the Kirby Institute prior to commencement of data collection. Within the interviews, participants were encouraged to reflect upon their motivations for advocacy and detail their personal experience. They were also asked to discuss their understanding of professional role obligations. Questions and prompts were delivered by the interviewer from the study interview domains. Informed written or verbal consent was obtained from all participants. For face-to-face interviews, a hard copy of the consent form was signed prior to interview commencement. For telephone interviews, verbal consent was formally obtained and audio-recorded at the beginning of the interview. Each participant was also provided with an information document and withdrawal of participation form during recruitment. There were no repeat or follow-up interviews.

Face-to-face interviews were conducted in the workplaces of participants or at their chosen quiet, public location with no other people present. Each interview was audio-recorded and manually transcribed. Interviews ranged from 27 to 150 minutes and averaged 55 minutes. Transcribed interviews were stored securely at the Kirby Institute using NVivo 11 software for management, and identifying details were removed. Field notes were used for discussion within the research team on a weekly basis, but not formally analysed as part of the data set.

Participants who expressed concern over anonymity were offered their interview transcript for deletions of identifying data. This was not for the express purpose of 'member checking' but to ensure that processes for anonymising the participant were adequate, transparent and acceptable to the individual involved.

**Table 2. Breakdown of sample statistics.**

| CHARACTERISTIC | SAMPLE (N = 32) | STUDY % |
|---|---|---|
| **INTERVIEW TYPE** | | |
| Face to face | 11 | 34.4 |
| Telephone | 21 | 65.6 |
| **GENDER** | | |
| Male | 13 | 40.6 |
| Female | 19 | 59.4 |
| **OCCUPATION** | | |
| General Practitioner | 7 | 21.9 |
| Paediatrician | 5 | 15.6 |
| Psychiatrist | 3 | 9.4 |
| Child psychiatrist | 3 | 9.4 |
| Medical student | 3 | 9.4 |
| Psychologist | 2 | 6.3 |
| Surgeon | 2 | 6.3 |
| Specialist physician | 2 | 6.3 |
| Junior doctor | 1 | 3.1 |
| Paediatric clinical nurse | 1 | 3.1 |
| Paediatric neurologist | 1 | 3.1 |
| Psychiatric nurse | 1 | 3.1 |
| Registered nurse | 1 | 3.1 |
| **ACADEMIC RANKING** | | |
| Professor | 6 | 18.8 |
| Associate professor | 5 | 15.6 |
| Senior lecturer[5] | 4 | 12.5 |

[5] If not already listed as a Professor or Associate professor.

## Data analysis

Qualitative data was thematically categorised. The iterative categorisation method (Fig 1) was used to identify thematic and conceptual similarities and differences between participants [67]. Coding in NVivo 11 was both inductive and deductive: an initial coding framework was developed from the interview question domains to guide analysis, and new codes were generated with successive transcripts. E.g. 'Personal Reputation.' As new codes were generated, they influenced structuring of subsequent interviews, in some cases with added questions. E.g. "Do you think there is a fear among some doctors of being branded as an activist?" Codes were refined on successive returns to the data until they represented a core set of interrelated themes. All data was coded by the first author and coding processes and the generation of themes was discussed within the research team at weekly meetings. We did not focus on achieving data saturation, but on ensuring that we included the broadest range of influential health care providers who were identified as active in the asylum seeker health rights movement.

## Results

A number of initial themes were generated from the data. Of these, four were found to be meaningful in addressing research aims and were refined as major themes. The first theme elucidates participants' professional and personal ethics; the second shows how these were challenged by immigration detention; the third provides insight into initial motivations (part 1)

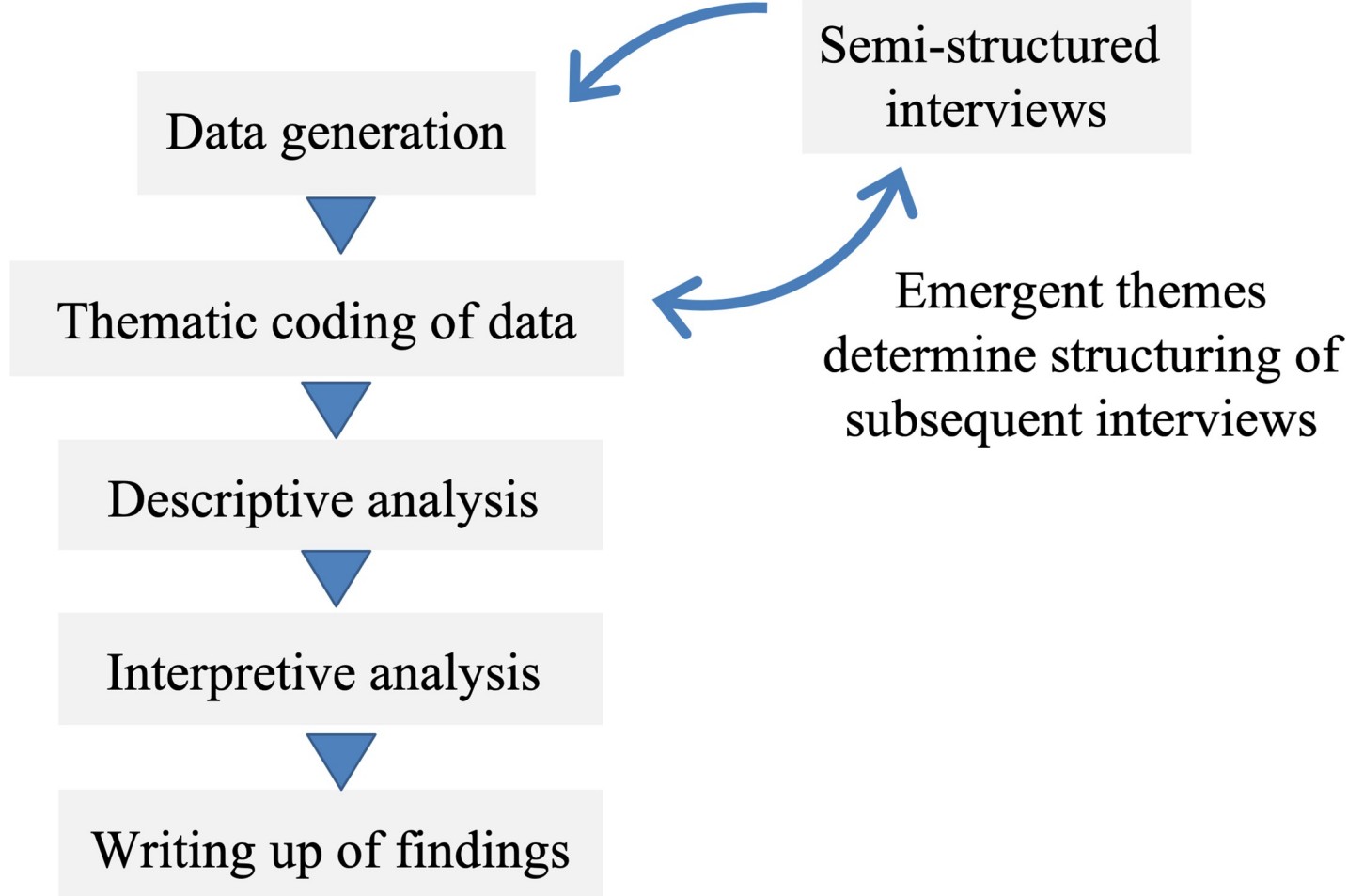

**Fig 1. Thematic analysis using iterative categorisation.** Fig 1 depicts the iterative categorisation method of data analysis, described by Neale [67] and Braun and Clarke [61], that was used in this study. The figure was constructed by the author.

for advocacy engagement, and longer term perseverance (part 2) with advocacy engagement; and the fourth illustrates participants' perceptions on the direction (part 1) and scope (part 2) of medical advocacy. The divisions into two parts of theme 3 and 4 reflect important delineations identified in participant' responses that added to the depth of existing themes, rather than creating new ones. These themes are presented below. Advocacy responses included a wide range of clinical and non-clinical actions, outlined in Fig 2.

## Theme 1—Personal ethics and perspectives on role obligations

### Patient-centred care

Respondents expressed strongly that *'the priority of medical professionals is looking after the patients' (P11)*. This duty of care was held to have primacy over all other contractual, legal or moral obligations: *'medicine has a higher calling to us than anything else' (P10)*.

### Social determinants of health

Holistic healthcare and engagement with social determinants of health was also described as being fundamental to medical professionalism. As P28 related: *'I don't see how you can do any*

**Advocacy responses**

## *Clinical*

**Individual patient advocacy**
*Such as refusing to discharge asylum seekers back into detention*

**Medico-legal and clinical reports**
*To assist with legally enforced medical evacuations*

**Clinical assessments**
*Either in person or via teleconference. Usually with independent health assessment teams. E.g. Royal Commissions, the medical evacuation response group, Doctors for Refugees or Kids Off Nauru*

**Specialist advice**
*E.g. as consultants to the Australian Human Rights Commission, or as government advisors to the DIBP*

**Asylum seeker health volunteer groups**
*Such as ASRC and refugee clinics*

**Individual patient advocacy**
*Engaging in acts of subversion to improve patient wellbeing*

**Complaints through internal channels**
*E.g. Christmas Islands Officer's Letter of Concern*

## *Non-clinical*

**Individual patient advocacy**
*Such as helping navigate health, disability, and monetary assistance programs, sponsoring a family's resettlement, and coordinating with schools*

**Whistleblowing**
*Breaking IHMS contracts to expose conditions*

**Research, position statements and publications**
*Publishing research results that oppose DIBP practices that aggravate ill-health, as well as opinion pieces, editorials and position statements from the AMA and colleges that advocate for asylum seeker health*

**Protests, campaigns and marches**
*Such as Kids Off Nauru campaign and Detention Harms Health march. Working with the media and engaging the public to enforce political action*

**Developing Medevac legislation**
*Promoting asylum seeker health in parliament*

**Letter writing and lobbying**
*Publicly and to local MPs and politicians*

**Public advocacy**
*Spreading awareness through interviews, panel discussions and public speaking in the media and at conferences*

**Witness testimony**
*In legal trials or in hearings for the Royal Commission*

**Witnessing and documenting**
*For future testimony*

**Key**
External to the ID system
Within the ID system

**Fig 2. Respondents' advocacy activities for the health of detained asylum seekers.**

*medicine without considering the sort of bio-psychosocial-cultural context within which things happen.'* Respondents expressed shared opinions that 'being a good doctor' meant addressing social-conditions that impede health and moving beyond insular disease-orientated clinical

practices to a big-picture approach. In this a number identified self-difference; relaying that theirs was *'a minority view' (P06)* amongst the profession. As one surgeon pointed out: *'There're many doctors who approach medicine quite differently, who see their obligations as only to the patient that is immediately in front of them' (P06)*. Some attributed this to a narrow focus in the medical teaching curriculum.

## Civic mindedness: Giving back, pro-bono work and social justice

Understandings of social factors of health were related to participants' advocacy actions, many of whom work with disadvantaged populations. Most respondents valued *'pro bono work [as being] important,'* and engaged because they *'had some skills to offer' (P06)*. For our participants, being a health professional is a privilege that comes with responsibilities. *'The privilege of knowing more and understanding how social conditions affect a person's health' (P20)* and the privilege of being *'in a position where we can make a difference' (P04)* confers a professional duty to engage in social-justice and advocacy. As P25 stated:

> *'You can't be insular in your profession. . . being a responsible citizen means using your professional knowledge and ability to help outside of your own clinic. . . And I think we have a duty as citizens and as doctors to be involved in those things.'*

More generally, an awareness *'of being fortunate,' (P28)* was a motivator for advocacy. Participants expressed beliefs that those *'in more fortunate situations do have a responsibility to help those who are in need' (P29)* and that they themselves have *'a responsibility both personal and professional, to give back in some way' (P31)*.

## Advocacy: A personal and professional duty

Strong beliefs were expressed across the board that, *'being an advocate is part of the role of being a health practitioner' (P14)*. When asked if their advocacy actions stemmed from personal ideals or professional responsibilities, participants expressed an inability to distinguish between the two. Over time, personal and professional morals become *'intertwined' (P31, P18, P26)*. Resultingly, our respondents *'wouldn't see [their advocacy actions] as ever being separate from [their] professional roles' (P28)*. For many, understandings of *'a duty to advocate that is part of our medical ethics,' (P10)* translated into a *'personal desire and a professional desire to advocate' (P04)*.

## Theme 2—Immigration detention: A challenge to role obligations

Professional role obligations were *'violated'* (P02) by respondents' experiences of immigration detention practices. For many, such challenges to professional ethics proved defining moments on a personal journey to becoming medical advocates.

## Initial shock and outrage

> *'It was like being punched, repeatedly. . . being there. It was just shock after shock. . . I'm not easily shocked. . . [but] nothing prepared me for this' (P03)*.

Those who had worked within immigration detention, both onshore and offshore, described an overwhelming awareness of how the detention environment was aggravating ill-health: *'You're dealing with patients who are suffering as a result of detention' (P10)*. Practicing within a bio-psychosocial framework, respondents *'couldn't think about these individuals without*

*thinking about the context and the environment,'* and described the futility of trying to improve health outcomes without first removing patients from IDCs: *'. . .you couldn't really make any difference, because the environment was continuing to be traumatising' (P28).*

Respondents found that their ability to provide clinical care was limited by substandard health resources and deliberate intervention of the DIBP refusing medical transfers. As P24 described:

*'You couldn't get them to see a paediatrician. You couldn't get them off the island. You couldn't get any ultrasounds done. You couldn't get stuff done. It was just ridiculous. . . [it was] impossible to treat to Australian standards.'*

A sense of professional outrage was reported when expert medical advice was superseded by DIBP security concerns in health matters. As a child health specialist relayed:

*'I assumed that I had some power. . . because in most other settings, you know, you're acknowledged as having expertise. . . And there's kind of been a respect for that expertise and a request for it. . .. Everywhere except in relation to immigration' (P28).*

Working within the system, the centrality of patient advocacy in medical professionalism was often denied. Senior health director P01 described role tensions between those clinicians who viewed advocacy as *'an intrinsic part of the role'* and the immigration department, whose understandings of this *'[were] very, very poor.'* He stated that advocacy was *'specifically restricted,'* and that *'any staff members who were seen to be advocating, were usually dealt with by disciplinary sort of interventions. . . sometimes they'd just sacked, outright, on the spot' (P01).* Instead of patient-centred care, respondents felt that *'IHMS['s] duty was to the government'* and they were *'not acting in the best interests of the patient' (P14).* Those in directorship roles described attempts to defend advocacy positions by *'pushing'* ethical guidelines *'back onto the department.'* But were *'rebuffed' (P01).* Of note, ethical codes were perceived to hold value as tools for communication with non-medical groups and government and were thus protective more than directive for our clinicians.

## Mandatory reporting and losing trust in the system

Role conflicts arose acutely around patients who had been abused in IDCs. Respondents described legal and professional obligations to report child abuse and assist in protection measures that were deeply aligned with personal morals; and were dismayed at being forced to *'[send their] patients back into an environment where they're going to be abused again' (P13).* Participant narratives suggested a loss of trust in the systems' protection measures and reporting pathways:

*'I had no faith that those people that I had a duty of care for were going to be kept safe. . . [and] I felt really outraged, and just despair. . . [that] I couldn't rely on the system to help them' (P10).*

And describe parallel 'internal to external' advocacy journeys:

*'We felt compelled to act. We initially attempted to advocate within IHMS. . . And then, that failed, and we began advocating more broadly after that' (P10).*

### Guilt, complicity and working within the system

Disenchantment with immigration detention as an appropriate environment for healthcare provision meant respondents were confronted with ethical conflicts of how to best fulfil central role obligations. Mental health clinician P32 chose to *'hid[e their] political views'* so they could remain working internally, because they *'didn't really want to leave [their patients] prematurely.'* Describing an 'innocence to experience' journey shared by many respondents, P32 related that initially *'I went naively'* but began to feel, *'a growing awareness all the time I was there. . . that I was a part of something that I oughtn't be a part of.'* Ultimately, *'I did see myself as complicit in the end.'* A number of clinicians similarly described feelings of responsibility, guilt and complicity.

For all participants, working within the immigration detention system became incompatible with personal and professional understandings of what it means to be a doctor.

> *'I was just completely flabbergasted. . . as to why any doctor would think, or any nurse or any health professional, would think that it was okay to be doing what we were doing' (P24).*

### Theme 3, part 1—Getting involved: Reasons for advocacy engagement

Respondent narratives of taking up advocacy roles reflected three integral stimuli: proximity, readiness, and personal ethics. The presence and union of all of these factors necessitated participants' advocacy responses (Fig 3).

### Becoming an advocate: Proximity

Proximity refers to participants' awareness of, and nearness to, perceived injustices and health-inequities of detained asylum seekers. For many, this came from experiences in immigration detention; either as health professionals or as refugees themselves. While proximity facilitates advocacy, the inverse is true for distance: *'it's hard when you haven't directly seen things' (P14)*. For others, personal involvement originated from their networks, through requests to do medico-legal clinical assessments or engage in asylum seeker health groups.

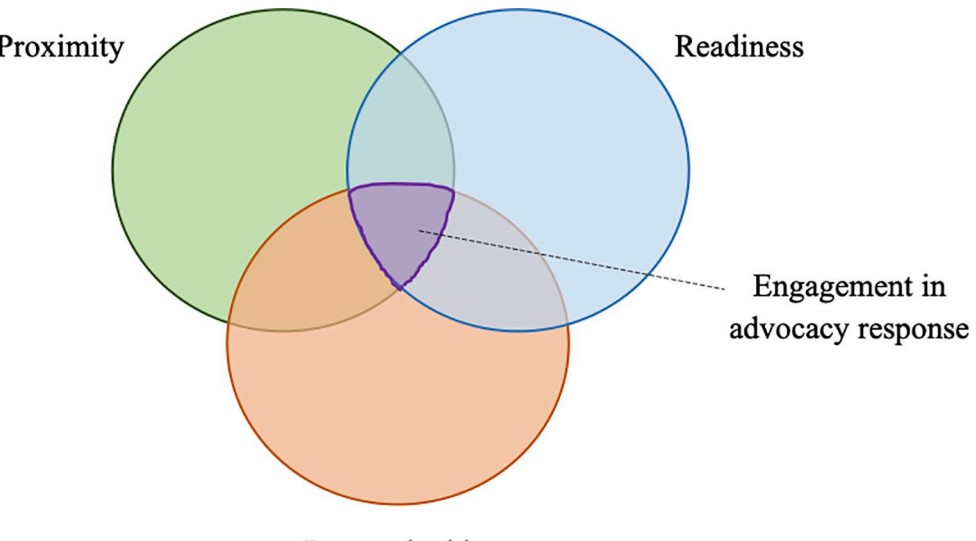

**Fig 3. Integral stimuli in becoming an advocate against mandatory immigration detention.**

Once involved, participants found that they *'started becoming aware' (P02)* and couldn't turn away: *'it's like a burning kind of concern, basically, that kind-of keeps eating me up (P08).*

All respondents relayed personal moments of 'serendipity' that precipitated involvement: receiving a call *'out of the blue' (P13, P06)*, or stumbling across an employment opportunity.

### Becoming an advocate: Readiness

Readiness denotes participants' temporal and individual circumstances that allowed them the capacity to advocate. All respondents for example, were already citizens or permanent residents of Australia. A senior clinician ruminated that:

> *'I'm now… very financially secure. So, would I have done this at an earlier stage in my life when I was trying to see three kids through and pay school fees? I wouldn't have quite gone the whole hog, put it that way.' (P25)*

Of note, self-reflections such as this imply that all three stimuli are necessary for action. With the absence of any one stimulus, two alone are not enough to necessitate an engagement in advocacy responses. In terms of personal readiness, if risks to job-security, financial stability or personal safety were deemed to be too high then individuals would not advocate. For P13, this occurred with the risk of imprisonment imposed by the 2015 Border Force Act: *'I had a five and six-year-old… Breaking the contract was one thing, but this was something very different… I stopped everything.'*

The Border Force Act (2015) was reported as having *'a huge chilling effect' (P10)* upon respondents engaged in advocacy actions at the time. The Act was passed with bipartisan support on July 1st, 2015 and made the disclosure of information obtained whilst working in Australian immigration detention punishable by up to two years of imprisonment. This was met with outrage from the medical community and a high court challenge was brought against the government. By September 2016 the Act was amended to exempt clinicians from the secrecy provisions [11].

### Becoming an advocate: Personal ethics

Respondents' personal ethics, as inseparable from core professional responsibilities (refer to *Advocacy: a personal and professional duty*), are integral for advocacy engagement. In the direction of their involvement (Fig 4), participants can be split broadly into two categories: accidental advocates (n = 12), who never would have seen themselves as engaging in advocacy roles, or jigsaw advocates (n = 20). The term jigsaw advocate describes participants who found that engaging in medical advocacy for asylum seekers was the final piece of a personal puzzle that comprised pursuing social justice and humanitarian endeavours. These respondents found that medical advocacy answered personal motivations for promoting social justice, and that the advocate role slotted well into their perceptions of self. For many, both categories are somewhat applicable: participants were inclined to social justice roles but were stung into asylum seeker advocacy by the factor of proximity. If understood as a spectrum of self-reported likelihood to have ended up as an advocate, most participants were near the middle (Fig 4), with a few at each extreme.

## Theme 3, part 2—Staying involved: Reasons for advocacy engagement

### Personal satisfaction

Participants described feeling gratified and fulfilled by their advocacy work: *'it makes you feel good' (P13).* *'[It]'s what, sort of, life's all about' (P25).* One participant working privately in a wealthy area identified *'a need within [them]self to do something like this.'* After treating the

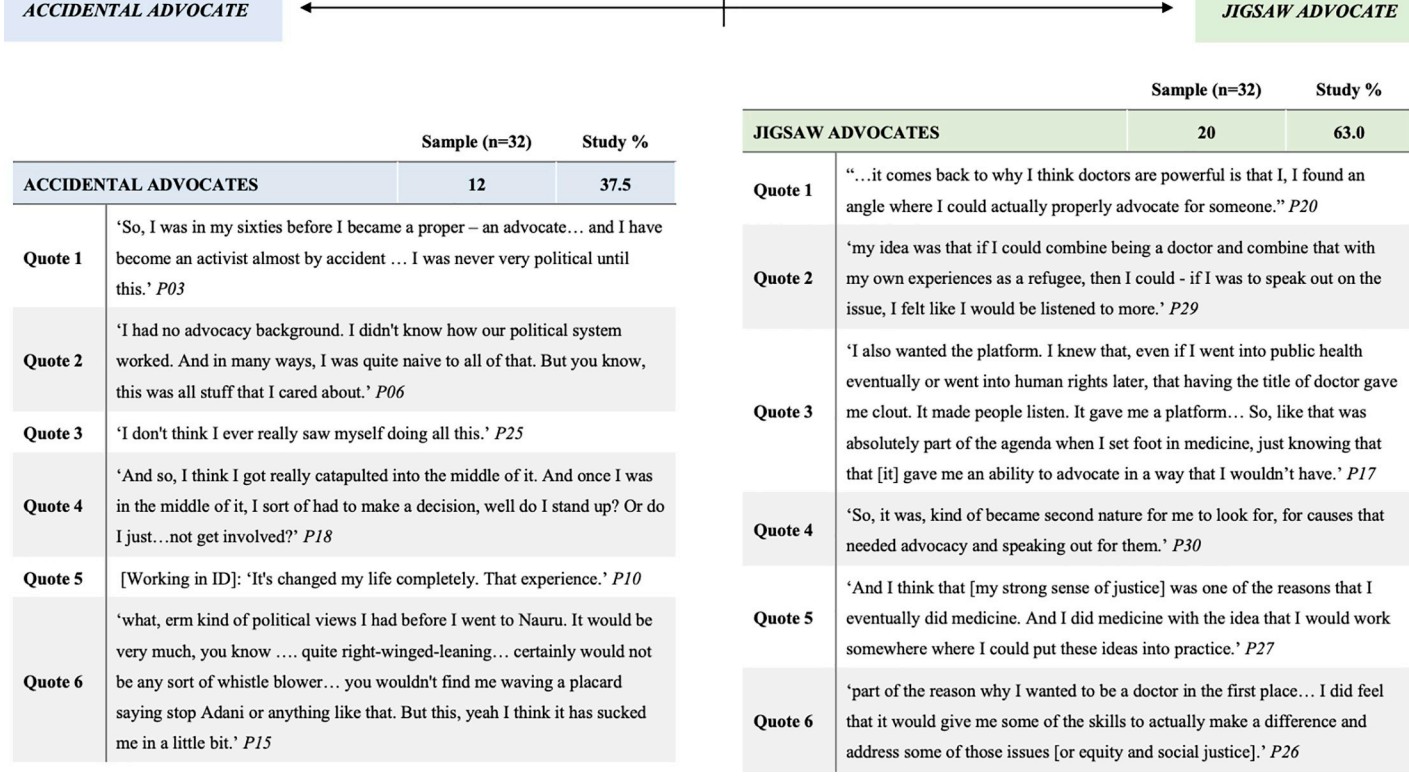

**Fig 4. Directional engagement of participants with illustrative quotes.**

'worried well' in their clinic, advocacy work was 'almost a relief' (P25). Others found advocacy rewarding because of the academic 'interest. . . opportunity and challenge' (P04) it provides.

## Self-preservation

In many cases, strong consciences meant that advocacy was in participants' best interests: 'the guilt of not saying something would have cost me more sleepless nights than the stress involved with saying something' (P18).

Advocacy was reported to be protective against the 'moral injury' (P10) of participants' experiences, as 'doing something means you don't just become a victim yourself' (P28). In this way, it helps counter 'feeling[s] of impotence and being denigrated and being powerless alongside the people that you're trying to help' (P28).

Self-preservation from future reprisal or judgement also proved incentivising. P15 stated that: 'I did think at one point: no, this is too much if I don't speak out because. . . when there's a royal commission down the line. . . we'd all be in the dock' (P15). Participants reported fearing that they would be on the wrong side of history, that: 'Nuremburg will happen one day and. . . I'm gonna be one of the perpetrators' (P16). For those participants with South African (n = 2) or Jewish (n = 4) backgrounds, this was a strong motivator: 'of course, I felt for the refugees but the, the most, the strongest feeling was of contamination with an evil process. . . I just couldn't do it' (P16).

## Personal ethics

Ultimately, advocacy answered the moral drives of participants. 'If I see something and I think it's wrong, it's really hard for me to just turn away. . .. it rubs on my conscience a bit' (P21). All

respondents held that, *'whistleblowing is a good thing' (P01)* and that there is *'a responsibility to speak out' (P29)*.

## Theme 4, part 1—Advocacy as a social movement: Future directions

Reflecting on advocacy as a means to effect change; participants identified collaboration and cohesion as essential issues. '*The most power comes when there's working together' (P28)*.

### Alliance building

Cross-sector collaboration was centralised in discussions around the future of asylum seeker advocacy movements, moving away from advocacy as tethered to the clinical sphere: *'all those groups; teachers and doctors and social workers. . . they're gonna drive change' (P22)*. In particular, respondents noted the value of working with lawyers for patient advocacy against oppositional government. Some participants stressed the need to engage asylum seekers as autonomous subjects in their own right: *'witnessing to them in a personal sense' (P28)* and gaining their active participation in any research or advocacy. To force policy change *'there needs to be a collective voice' (P18)*, as individuals too easily *'get pigeonholed' (P18)* or denigrated as *'trouble-making activist[s]' (P02)*. Speaking from experience with the Medevac and marriage-equality campaigns, P30 said: '*the more that you can work together for a common cause, and have a single message that's going out, the more powerful your political position will be.'*

The Medevac campaign was cited by respondents as a successful advocacy movement achieved through collaboration and alliance building. The Medevac legislation was introduced to parliament in late 2018 and passed into law on the 1st March 2019. It allowed for the urgent medical transfer of asylum seekers from offshore processing centres (typically on Nauru island or Papua New Guinea) to Australia for medical treatment and / or assessment [68]. Of note, the legislation was effective in its objectives but short lived. A bill repealing the Medevac legislation was put forward by the Home Affairs Minister Peter Dutton on behalf of the Coalition government in July 2019. This bill passed the lower house on July 24th and the senate on December 4th 2019 and the medical transfer provisions of the Medevac legislation were removed from the *Migration Act 1958* [69, 70].

### Collegial backing and support

Having the backing of professional bodies was recognised as lending 'clout' and 'credibility' to such messages, and a number of participants looked for *'strong leadership from the AMA' (P15)*. There was some disagreement here; while many held that the AMA could empower and unify advocacy for asylum seeker health, others were disenchanted by what they considered *'a watering-down' (P28)* of the political stance. Both P01 and P06 noted that the AMA is primarily a *'doctor's union,'* whose priority will always be *'the welfare of doctors in this country.'* Further, having a *'de-facto voice of the medical community' (P06)* may discourage individuals from speaking out. Others pointed out that *'doctors are a multifaceted people' (P17)* with a *'huge range of attitudes' (P10)*, thus making a consensus on advocacy-direction difficult.

Respondents did agree, however, that having the backing of ones' profession was essential as a means of support. '*How do you keep doing the work? I think you need colleagues. You need professional support, like, from your professional bodies. You can't do it on your own' (P28)*. Such support was perceived as being protective against professional risks and adverse impacts of advocacy on personal wellbeing. Those who experienced hostility from their hospital management, or were reprimanded for advocacy actions, reported negative consequences: *'we were*

*completely burned out, and just completely traumatised. . . it was awful' (P18). '[It's] lonely if your hospital and your institution doesn't support you' (P04).*

Of note, participants reported that negative reactions from hospitals and state health organisations were likely due to concerns over funding and constraints of the public health system. *'This is what [the hospital executive] said: we really admire the work you're doing. But don't ever mention the hospital when you're speaking publicly. . .. because you know, the Liberal government fund our hospital. And this could affect the funding of all the other services.' (P13)*

Peer networks and teams with a *'culture of advocacy' (P17)* were highly valued by participants as both protective and empowering: *'strength and safety in numbers. . . really empowers you to feel like you can speak up and advocate' (P17).* Furthermore, clinicians relied on peers for guidance: *'I consulted colleagues that I trusted more than the guidelines' (P23),* indicating the importance of shared ethical understandings in an interdependent profession.

## Theme 4, part 2—Defining medical advocacy and its scope in systems of abuse

There was a consensus among respondents that advocacy is more powerful when it stems from professional expertise, and that moving outside of this sphere risks credibility. As P14 stated: *'it's important that we are not the people chaining ourselves to buildings and stuff, because. . . that is not what will have an impact. What will have an impact is just medical practitioners reporting on the medical facts of the case.'*

Resultingly, in their advocacy respondents remained *'very focused on medical stuff,'* as anything else *'will detract from our message, which is: we're doctors, and we're concerned about healthcare' (P02).* This attitude indicates a tacit understanding held by respondents about the role of medical advocates: specifically, that medical advocacy is necessarily distinct from civic virtue and must come from a strictly health orientated outlook. Respondents felt that their role as medical advocates required a level of political involvement necessitated by the adversarial position of the Australian government and the politicised nature of immigration detention: *'we can't be politically neutral when the politics is harming the people' (P03).* However, separating personal political opinions from medical advocacy on a politicised issue was held to be an essential, and sometimes difficult, part of maintaining professionalism in advocacy.

Most participants indicated that outside of individual clinical advocacy, *'what the role of doctors is in advocating is very unclear, particularly in the public and in the media' (P04).* Furthermore, participants felt that *'we need to make a distinction between protest and advocacy (P23).* There was a general call for clarification from professional medical bodies, as well as for specific guidelines on the role of medical advocacy and whistleblowing under policies antithetical to health.

> *'If you can encourage people to speak out [then] you should give them a decent framework to say. . . this is your line in the sand' (P15).*

## Discussion

This study explored the perspectives of Australian medical and healthcare professionals advocating for the health of refugees and asylum seekers in mandatory immigration detention. Our findings provide insight into factors that motivate clinician engagement and perseverance with medical advocacy, specifically in systems where patients' health rights are perceived to be infringed. Results indicate that advocacy responses originate from the union of three stimuli: personal ethics, proximity and readiness (Fig 3). These findings are important because medical

advocacy is a critical element for addressing human rights violations in immigration detention. In line with the aims of empirical bioethics to 'bring about ethically defensible changes to practice' [58], we suggest that each of these three factors should be addressed in future endeavours to promote advocacy within the profession.

Regarding the first factor of personal ethics, respondents reported that there is no line between their personal and professional ethics. This illustrates a widely-accepted position that doctors' professional identities become an integral part of their personhood [71, 72] and goes some way in explaining the depth of participants' responses to role violations in the immigration detention system. When professional obligations are deeply aligned with personal morals, compromising professional integrity constitutes a moral injury. Respondents turned to advocacy as the only available action not antithetical to their personal ethics. This finding is significant, as it lends support to the theory that medical advocates act primarily from personal ideals [2, 6]. Further, that nearly two-thirds of respondents were jigsaw advocates, suggests that civic-mindedness is an important influencer in decisions to take up advocacy roles. If medical schools view advocacy as being desirable in their graduates, then admissions processes should thus consider dispositions for social-responsibility in applicants [1, 4].

The finding of proximity as another integral factor in respondents' advocacy engagement corroborates results of previous empirical studies [2, 4] and supports growing evidence that exposure to health inequities may encourage physician advocacy [73]. We suggest a number of ways in which increased proximity may be applied to this effect.

Firstly, there is an opportunity in medical schools. This finding offers guidance to medical educators attempting to foster advocacy as a professional competency in graduates. Medical training programs could:

1. sponsor supported places for students with lived proximity to health inequities, including those from disadvantaged areas; and

2. make social justice contribution an imperative for qualification.

In implementing this latter, our findings support recommendations for requiring structured clinical placements in marginalised communities [1, 2, 4, 74] so as to facilitate proximity.

Such action should be taken alongside existing coursework on social determinants of health. Our participants reported feeling compelled to act when they perceived the health of detainees to be directly at risk, and experiences of compromised patient care in ID was common in driving an advocacy response. Clarifying evidence for causative links between broad determinants and ill-health may thus promote clinician involvement in advocacy roles. This supports findings by Gruen et al. [6], who describe how the perceived importance physicians assign to advocacy on any particular issue is 'related to how directly the issue concerns individual patients' health.'

Secondly, peer networks are significant. Exposure through peer-networks may also prove an essential medium for simulating clinician proximity. Our findings around clinician interdependence, and perspectives on the protective and empowering nature of teamwork, lends support to observations that doctors derive sustenance from advocacy as a team activity [75]. Accordingly, reconceptualisation of medical advocacy as a group activity may encourage increased engagement among the profession [2] and mitigate fears around legal, reputational and professional ramifications.

This finding is particularly important in light of current bioethics literature on Australian immigration detention: which primarily focuses on informing clinician individualism and uses ethical discourses to recommend application of physician values [34, 76]. Based on

participants' insights, a better approach might centre future bioethical discussions around methods for a collaborative and cohesive advocacy response, essential for social and political reform.

Lastly, physician advocacy needs to be normalised through policy and leadership. Identifications of self-difference among study participants underscore sentiments in the literature that 'successful physician advocacy tends to be exceptional' [1]. To the degree that the medical profession truly considers advocacy a professional commitment, action must be taken to reduce its exceptionalism. In pursuing this aim, Rothman [74] suggests that professional and board-certifying organisations could require doctors to perform service-work with marginalised groups to maintain certification. For our respondents, this would formalise what is already perceived to be a professional duty to engage in social justice and advocacy. If successfully implemented, such a policy would foreground and normalise the shared commitment of medical professionals in embracing advocacy as a central role-obligation.

The issue of readiness is best addressed by ensuring the safety of clinicians who advocate. For our respondents, the perceived risks involved with medical advocacy dictated their capacity to engage in advocacy responses. The major protective factors against these risks included peer-support and collegial backing. Revealingly, narratives describing hostility and reprimand from hospital management suggest that this essential support is not always provided to whistleblowers and advocates in the health system, constituting a major barrier to advocacy. We conclude that in order to promote medical advocacy, the profession of medicine must take decisive action in encouraging and protecting its whistleblowers.

Promoting whistleblowing in healthcare has been somewhat addressed in bioethics literature. Both Faunce [77] and Palmer and Rogers [78] seek to address this current professional weakness through pedagogy; offering different approaches to the formalised inclusion of health-whistleblowing in medical curricula. For effective change, however, such efforts to influence normative moral theory must be coupled with practical and enforceable measures [36]. As Faunce and Bolsin [79] point out, Australian accreditation-organisations and health systems currently deflect whistleblower criticisms and 'actively suppress the positive institutional culture of open disclosure.' Whistleblower protection schemes for hospitals and health-organisations would need to be enforced by regularity authorities [36], and any perceived adverse effects of advocacy engagement (for example, on funding and access to government) would need to be mitigated to increase adherence. Here, organisational bodies could look to the literature on NGO health-advocacy; which proposes effective tactics managing oppositional advocacy with partnership approaches to government [80].

Recently, the AMA has proposed reform to sections of the Medical Board of Australia's ethical guidelines [3] that impinge upon medical advocacy. Specifically, they recommend removing statements that could be interpreted as 'trying to control what doctors say in the public arena' and as 'coerc[ing] doctors into complying with relevant laws that are inconsistent with professionally accepted standards of medical ethics' [81]. While applying these recommendations may go some way in formalising protections for Australian medical advocates, neither body has produced any explicit guidelines promoting whistleblowing or protecting whistleblowers. Doing so would be an essential first step in raising the visibility of advocacy as a professional imperative.

The scope and practice of medical advocacy remains the cause of debate among the medical community [6]. While many argue for the extension of medical advocacy to the public sphere [1, 37, 77, 82], others hold that politics has no place in medicine, and that engagement in social justice will necessarily displace clinical duties [83, 84]. This debate is reframed in the setting of Australian immigration detention, with respondents describing how the most fundamental professional responsibilities can become politicised, and how medical advocacy can become essential for the delivery of standard clinical care. These findings are not novel and have been reported in observational literature from clinicians working in IDCs across the world [10]. In systems that diminish

patients' health rights, medical advocates play a crucial role in documenting human rights abuses [10, 30, 31, 34, 38] and countering government claims of care equivalence [13, 17]: actions that lie far beyond the scope of traditional clinical medicine. Results of respondents' delineation of medical advocacy in the political realm are thus illuminating and could provide guidance to clinicians attempting to define or engage in medical advocacy worldwide.

## Limitations and further research

The flexibility of in-depth interviews allowed participants to frame their experiences in their own terms. This, together with the relatively large sample size for a qualitative study, is a study strength. A limitation of this study was that the perspectives of non-medical advocates, and of medical professionals who did not take up advocacy roles, were not accounted for. Further, discussions were mostly focused upon medical advocacy relating to doctors—to the exclusion of other health professions. The role of health advocacy among nurses, psychologists and allied health professionals is a direction for future research.

Another limitation was the absence of asylum seekers perspectives. While two respondents had previously been asylum seekers in IDCs, their understandings are informed by their identity as medical professionals and are thus not applicable as patient perspectives. Patient input is important for informing medical professionalism [8] and reinstating subject dialogue in asylum seeker research is essential for ethical conduct [85]. While interviewing detained asylum seekers was outside the scope and the ability of this study, their perspectives would have enriched the data. Relatedly, we call for increased researcher access into Australian IDCs.

The negative consequences of advocacy on personal wellbeing was an unanticipated theme that we identified. While also outside the scope of this study, investigation into the prevalence of vicarious trauma and compassion-fatigue among medical advocates, and the specific factors that may put them at risk, is a direction for future research.

## Conclusion

This study offers a unique account of the perspectives of Australian medical professionals advocating for the health of detained asylum seekers. Our findings show that three integral stimuli are needed for clinician engagement in advocacy, offering important guidance for policy-makers, medical leaders and educators attempting to promote professionalism in medicine. Considering current global expansions of protectionist politics, we suggest that this research into the role of physician advocates in systems that diminish patients' health rights, and how best to support these advocates, is a timely and important resource. We hope that this study adds to a stronger evidence base for the practice of advocacy in medicine; a practice that is essential for safeguarding patients' universal right to health.

## Supporting information

**S1 Checklist.**
(PDF)

**S1 Table. Results—illustrative quotes.**
(DOCX)

## Acknowledgments

I would like to thank the medical advocates who participated in this study, and Brock Sherlock for his patience and help transcribing interviews.

## Author Contributions

**Conceptualization:** Rohanna Stoddart, Bridget Haire.

**Data curation:** Rohanna Stoddart.

**Formal analysis:** Rohanna Stoddart.

**Investigation:** Rohanna Stoddart.

**Supervision:** Paul Simpson, Bridget Haire.

**Writing – original draft:** Rohanna Stoddart.

**Writing – review & editing:** Rohanna Stoddart, Paul Simpson, Bridget Haire.

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
