## [Decision Letter · Decision Letter 0]

3 Feb 2020

PONE-D-19-29972

Advocacy in Medicine

PLOS ONE

Dear Miss Stoddart,

Thank you for submitting your manuscript to PLOS ONE. After careful consideration, we feel that it has merit but does not fully meet PLOS ONE’s publication criteria as it currently stands. Therefore, we invite you to submit a revised version of the manuscript that addresses the points raised during the review process.

The manuscript has been assessed by two reviewers, their comments are available below.

The reviewers find the work of relevance but have raised a number of concerns that need attention in a revision. The reviewers request improvement to the reporting of the methodology, including additional information on the methodological orientation employed for the analyses. The reviewers also note that the research aims need to be more clearly articulated and the definition of medical professions employed in the study should be clarified.

Could you please revise the manuscript to address the items raised.

In addition to the items raised by the reviewers, please revise your title to ensure it aligns to the work that is reported, the title currently refers to advocacy in medicine in general terms, while the study had a major focus on the context of refugee and asylum-seeker populations, this must be made clear from the title.

We would appreciate receiving your revised manuscript by Mar 17 2020 11:59PM. Please include the following items when submitting your revised manuscript:

We look forward to receiving your revised manuscript.

Kind regards,

Iratxe Puebla

Deputy Editor-in-Chief, PLOS ONE

2. Please consider changing the title so as to meet our title format requirement (https://journals.plos.org/plosone/s/submission-guidelines). In particular, the title should be "Specific, descriptive, concise, and comprehensible to readers outside the field" and in this case it is not informative and specific about your study's scope and methodology (the title entered in your submission form just reads "Advocacy in Medicine".

3. In the ethics statement in the Methods and online submission information, please clarify whether consent was written or verbal.  If verbal, please also specify: 1) whether the ethics committee approved the verbal consent procedure, 2) why written consent could not be obtained, and 3) how verbal consent was recorded. If your study included minors, state whether you obtained consent from parents or guardians. If the need for consent or parental consent was waived by the ethics committee, please include this information.

4. Please remove your figures from within your manuscript file, leaving only the individual TIFF/EPS image files, uploaded separately.  These will be automatically included in the reviewers’ PDF.

Reviewers' comments:

Reviewer's Responses to Questions

**Comments to the Author**

1. Is the manuscript technically sound, and do the data support the conclusions?

Reviewer #1: Yes

Reviewer #2: Partly

2. Has the statistical analysis been performed appropriately and rigorously? 

Reviewer #1: N/A

Reviewer #2: No

3. Have the authors made all data underlying the findings in their manuscript fully available?

Reviewer #1: Yes

Reviewer #2: Yes

4. Is the manuscript presented in an intelligible fashion and written in standard English?

Reviewer #1: Yes

Reviewer #2: Yes

5. Review Comments to the Author

Reviewer #1: Manuscript Number: PONE-D-19-29972

Article Type: Research Article

Full Title: Advocacy in Medicine

Short Title: A study of medical professionals upholding medical ethics in the face of Australian immigration practices

Corresponding Author: Rohanna Isabelle Stoddart

University of New South Wales

Sydney, New South Wales AUSTRALIA

Reviewer comments

General

This is an impressive report addressing an area of critical importance to the health of refugees and asylum seekers detained under various governments’ policies. The authors’ achievement in collecting, organising, and presenting qualitative data garnered from a significant number of health care practitioners deserves high commendation. Their methods were appropriate to the task, clearly described, and executed effectively (by their description). Their work largely met their main objectives of exploring and analysing the experiences and perceptions of medical practitioners who advocate for detained refugees and asylum seekers. Their engagement with literature related to the topic is substantial, and their recommendations about medical education ring true.

Most importantly, the data collected and laid out for the reader are original and provide insights not previously available to scholars, so the work has great value on account of this alone.

My comments generally belong in the minor category, but one issue appears to be of greater significance.

Significant issue

It is clear from the background/rationale, discussion, and recommendations that the authors’ interest is in the medical profession, i.e., doctors/physicians. At various points, however, a more expansive definition of medical professional – encompassing a wide range of health professionals – is used. One area where this calls for greater clarification and, possibly, justification is in the use of a sample containing a wide range of different health professionals to support analysis, discussion, and conclusions that appear to focus on doctors.

Minor issues

Research aims (lines 97-103): the second part of the second aim is a very important and successful aspect of the report and thus might justify being stand-alone.

Lines 125-126: It would be helpful to the reader (especially if non-Australian) to have the role of IHMS and its contract with the Australian government more completely explained.

Paragraph at lines 133-142: It would be helpful to the readers’ acceptance of the literature summary to have previously introduced the reader to the ‘wrongdoings’ of DIBP and the systematic abuses referred to here.

Line 147: Clarify that Hippocrates belongs to the era BC.

Section on normative tools: This seems like a comprehensive list. Is it possible to clarify the basis for selecting the entries and to state whether it is intended to be exhaustive?

Figure 1 is effective at communicating the procedure.

Results, line 284: The reader would benefit from understanding the basis for your decision to present only 4 (presumably the most important) of the themes; perhaps the non-included themes could be listed for completeness.

I found the organisation of the results into four major themes convincing and coherent; advance flagging of the rationale for splitting each of themes 3 and 4 into two parts would be helpful.

Lines 615-623: The authors ought to be given the opportunity to update this section in light of recent decisions in the Australian parliament.

Paragraph at 719-731: Pro bono work is already a feature of medical work; are the authors convinced that changing priorities and shifting towards the legal model would be beneficial and feasible overall? Some scholars in this area, notably Thomas Huddle, are chary of demanding socially focused actions of doctors.

Conclusion

I enjoyed reading this work and consider its clear and thorough description of practitioners’ perspectives on ethical dimensions of work in immigration detention to be illuminating and important.

Reviewer #2: The study is of significant importance as it is an area of limited research, particularly the views of medical professionals. The findings have the potential to provide relevant information to the various stakeholders.

The introduction and background provides adequate information to the reader. The selected literature was appropriate and builds up to the study rationale. Part of the rationale included the importance of the “role of medical advocates” Line 183. Perhaps this should be included in the discussion. A paragraph will be adequate as the study findings does have the relevant information to discuss the role of medical advocates.

Line 191 “shape medical advocates” I understand what you mean, but maybe this should be expanded. Shape them in what way?

Also, the authors state “factors that shape” – be specific, for example motivational factors. It will link directly to your findings.

Line 193 “participants’ – suggest change to the medical professionals, as you are only referring to them, not students or the refugees themselves.

The aims need to clarified as the aims in the abstract are somewhat different from the aims in the study. I also suggest that the “Research aims” section be moved after the rationale.

The methodology section requires major revision. There are problematic assumptions regarding the methods and rigour. Firstly, the COREQ guidelines were used, which may also be debatable. Please refer to:

Hannes, K., Heyvaert, M., Slegers, K., Vandenbrande, S., & Van Nuland, M. (2015). Exploring the Potential for a Consolidated Standard for Reporting Guidelines for Qualitative Research: An Argument Delphi Approach. International Journal of Qualitative Methods. https://doi.org/10.1177/1609406915611528

Nevertheless, not all information is reported as per COREQ checklist within the manuscript. For e.g. the methodological orientation and theory - What methodological orientation was stated to underpin the study? e.g. grounded theory, discourse analysis, ethnography, phenomenology, content analysis. This not presented in the data. Furthermore, this orientation should be linked to the study findings in the discussion.

The data analysis, Line 269 “emergent themes” and the Line 275 “Braun and Clark” is controversial. Please read:

Braun, V., & Clarke, V. (2019). Reflecting on reflexive thematic analysis. Qualitative Research in Sport, Exercise and Health, 11(4), 589–597. doi:10.1080/2159676x.2019.1628806

This shows a shift in their (Braun and Clark) thinking from 2006, which poses major flaws in the analysis.

It is also recommended that the following paper be read: Smith, B., & McGannon, K. R. (2017). Developing rigor in qualitative research: problems and opportunities within sport and exercise psychology. International Review of Sport and Exercise Psychology, 11(1), 101–121. doi:10.1080/1750984x.2017.1317357

The paper needs to demonstrate rigour, hence demonstrating quality, qualitative data analysis.

The results appeared to be very long, although was easy to read and understand. The relevance of selected quotes is questionable. There are too many quotations. It is recommended that the results section be condensed. Furthermore, the sub-headings per theme, are these sub-themes? Please clarify.

As mentioned earlier, the discussion should link to the theoretical framework. Too many subheadings should be avoided in this section. Overall, the structure of the paper needs to be revised.

The place/country of study of the health professionals should be taken into account as this may have influenced their responses to questions. It is also relevant since the results are to influence the education of the medical professionals training in Australia. This could be a limitation of the study.

Technical considerations:

Asylum seekers vs asylum-seeker

Well being vs well-being

P02 vs P02 (italics)

The indentation of paragraphs-looks “messy”

Please check for consistency.

Line 583 “they’d” – this should be in full

6. PLOS authors have the option to publish the peer review history of their article (what does this mean?). If published, this will include your full peer review and any attached files.

Reviewer #1: No

Reviewer #2: No

---

## [Author Response · Author response to Decision Letter 0]

30 Mar 2020

Response to reviewers

We are very grateful for the reviews provided by the editor and each of the reviewers of this manuscript. 

We have made significant revisions to the ‘Methodology’, ‘Research aims’ and ‘Discussion’ sections of the manuscript. Most notably, we have expounded on the reporting of the methodology and included additional information on the conceptual framework that was used. We have also attached a supporting document that details where each item on the COREQ guidelines is addressed in the paper. The research aims and the definition of medical professions involved in the study have been clarified and a number of revisions have been made throughout in line with reviewers’ suggestions. 

Please see below our detailed response to each comment made.

EDITOR REVIEW: 

Editor comment 1 1. Please ensure that your manuscript meets PLOS ONE's style requirements, including those for file naming.

Authors’ response Some revisions were made to the title page and subheadings to ensure style requirements were being met. 

Editor comment 2 2. Please consider changing the title so as to meet our title format requirement (https://journals.plos.org/plosone/s/submission-guidelines). In particular, the title should be "Specific, descriptive, concise, and comprehensible to readers outside the field" and in this case it is not informative and specific about your study's scope and methodology (the title entered in your submission form just reads "Advocacy in Medicine"

Authors’ response We have changed the title to “Medical advocacy in the face of Australian immigration practices: a study of medical professionals defending the health rights of detained refugees and asylum seekers”.

The short title now reads “Medical advocacy for refugees and asylum seekers”.

Editor comment 3 3. In the ethics statement in the Methods and online submission information, please clarify whether consent was written or verbal. If verbal, please also specify: 1) whether the ethics committee approved the verbal consent procedure, 2) why written consent could not be obtained, and 3) how verbal consent was recorded. If your study included minors, state whether you obtained consent from parents or guardians. If the need for consent or parental consent was waived by the ethics committee, please include this information.

Authors’ response Apologies for not making this clearer in the manuscript and in the online submission information. This has been revised accordingly in both. 

To clarify, of the 32 participants interviewed: 29 signed written consent forms and 3 provided verbal consent. The 29 who provided written consent included every participant with a face-to-face interview. In all cases, participants were emailed a consent form and information document when email contact was established. Three participants interviewed by telephone underwent a formal, scripted consent process at the beginning of their interviews to obtain verbal consent. This was audio-recorded. A verbal consent option was approved by the HREC. We have included a copy of the verbal consent script in the supporting documents. 

Specifically: (1) the ethics committee approved the verbal consent procedure. (2) Written consent could not be obtained as participants were not seen face-to-face and failed to return written consent form after two reminders. (3) Verbal consent was audio-recorded at the beginning of interview. No minors were included in the study. 

Please note the revisions made to the manuscript concerning the above:

Line 336 - 337: “Approval from UNSW Sydney’s HREC was received for this study (HC180778) and included the use of both written and verbal consent.”

Line 420 – 423: “Informed written or verbal consent was obtained from all participants. For face-to-face interviews, a hard copy of the consent form was signed prior to interview commencement. For telephone interviews, verbal consent was formally obtained and audio-recorded at the beginning of the interview. Each participant was also provided with an information document and withdrawal of participation form during recruitment.”

Editor comment 4 4. Please remove your figures from within your manuscript file, leaving only the individual TIFF/EPS image files, uploaded separately. These will be automatically included in the reviewers’ PDF.

Authors’ response We have removed all figures from within the manuscript file and left the individual TIFF/EPS image files. 

REVIEWER 1: 

General: This is an impressive report addressing an area of critical importance to the health of refugees and asylum seekers detained under various governments’ policies. The authors’ achievement in collecting, organising, and presenting qualitative data garnered from a significant number of health care practitioners deserves high commendation. Their methods were appropriate to the task, clearly described, and executed effectively (by their description). Their work largely met their main objectives of exploring and analysing the experiences and perceptions of medical practitioners who advocate for detained refugees and asylum seekers. Their engagement with literature related to the topic is substantial, and their recommendations about medical education ring true.

Most importantly, the data collected and laid out for the reader are original and provide insights not previously available to scholars, so the work has great value on account of this alone.

My comments generally belong in the minor category, but one issue appears to be of greater significance.

R1 comment 1 1. Significant issue: It is clear from the background/rationale, discussion, and recommendations that the authors’ interest is in the medical profession, i.e., doctors/physicians. At various points, however, a more expansive definition of medical professional – encompassing a wide range of health professionals – is used. One area where this calls for greater clarification and, possibly, justification is in the use of a sample containing a wide range of different health professionals to support analysis, discussion, and conclusions that appear to focus on doctors.

Authors’ response We agree with the reviewer’s comment about the need for clarification and justification on this issue. 

While we felt it important not to exclude the voices of prominent health advocates, the vast majority of participants were medical doctors. Furthermore, the scope of the study would have had to be dramatically increased if we had attempted to separate out the health professions, likely to the detriment of the cohesiveness of the study as a whole. Unfortunately, this has led to the exclusion of other health professions in the analysis and discussion section. 

We have added the following to the ‘Participant recruitment’ section of the methodology: 

“It is important to note here that the study is focused mainly on medical doctors and medical training, to the exclusion of other health professions. We felt it was important to include the voices of medical advocates from all health professions who had published influential papers or were otherwise leading advocates for asylum seeker policy reform. The vast majority of participants (28 of 32), however, were medical doctors or students, and this is reflected in the results, discussion and conclusion." (Lines 383 – 388)

We have also mentioned this in the ‘Limitations’ section. 

“Further, discussions were mostly focused upon medical advocacy as related to doctors - to the exclusion of other health professions. The role of health advocacy among nurses, psychologists and allied health professionals is a direction for future research.” (Line 1205 – 1207). 

R1 comment 2 2. Research aims (lines 310 - 320): the second part of the second aim is a very important and successful aspect of the report and thus might justify being stand-alone.

Authors’ response Thank you for pointing this out, we agree and have separated this out as suggested: “1. To document the personal views and experiences of Australian medical professionals advocating for the health of refugee and asylum seekers populations in mandatory immigration detention” (Line 311). 

The research aims have also been clarified and moved to after the rationale of the study, lines 310 - 320, in answer to reviewer 2’s suggestions Please refer to R2 comment 4 authors’ response. 

R1 comment 3 3. Lines 125-126: It would be helpful to the reader (especially if non-Australian) to have the role of IHMS and its contract with the Australian government more completely explained.

Authors’ response In the ‘Background’ section of the manuscript we have added the section: “Healthcare delivery in offshore immigration detention: a systemic failing” (Lines 178 – 192). The paragraphs added here explain the role of both the IHMS and the DIBP in regard to Australian immigration detention and present a summary of the systemic abuses and wrongdoings of these organisations as related to comment 4 below. 

R1 comment 4 4. Paragraph at lines 133-142: It would be helpful to the readers’ acceptance of the literature summary to have previously introduced the reader to the ‘wrongdoings’ of DIBP and the systematic abuses referred to here.

Authors’ response Please see response R1 comment 3 above. 

R1 comment 5 5. Line 147: Clarify that Hippocrates belongs to the era BC.

Authors’ response This has been clarified in line 228. 

R1 comment 6 6. Section on normative tools: This seems like a comprehensive list. Is it possible to clarify the basis for selecting the entries and to state whether it is intended to be exhaustive?

6.1. Figure 1 is effective at communicating the procedure.

Authors’ response This has been clarified: “Not intended to be an exhaustive list, the instruments presented in Table 1 were chosen for their normative power and relevance to Australian clinicians.” (Lines 255 – 257). 

R1 comment 7 7. Results, line 284: The reader would benefit from understanding the basis for your decision to present only 4 (presumably the most important) of the themes; perhaps the non-included themes could be listed for completeness.

7.1. I found the organisation of the results into four major themes convincing and coherent; advance flagging of the rationale for splitting each of themes 3 and 4 into two parts would be helpful.

Authors’ response The explanation of the results has been clarified in lines 463 – 471. We found four themes that addressed our research aims and refined these as the major themes. The 20 or so non-included themes were initial themes generated from the data that were not considered to be sufficiently refined for inclusion, did not appropriately address the research aims, or were reworked into the existing major themes. We have flagged the splitting of themes 3 and 4 as suggested and provided a brief rationale. 

“A number of initial themes were generated from the data. Of these, four were found to be meaningful in addressing research aims and were refined as major themes. The first theme elucidates participants’ professional and personal ethics; the second shows how these were challenged by immigration detention; the third provides insight into initial motivations (part 1) for advocacy engagement, and longer term perseverance (part 2) with advocacy engagement; and the fourth illustrates participants’ perceptions on the direction (part 1) and scope (part 2) of medical advocacy. The divisions into two parts of theme 3 and 4 reflect important delineations identified in participant’ responses that added to the depth of existing themes, rather than creating new ones.” (Lines 463 – 471)

R1 comment 8 8. Lines 615-623: The authors ought to be given the opportunity to update this section in light of recent decisions in the Australian parliament.

Authors’ response This section has been updated, most notably detailing the successful repeal of the Medevac legislation on December 4th, 2019 in the senate (lines 901 – 902). 

R1 comment 9 9. Paragraph at 719-731: Pro bono work is already a feature of medical work; are the authors convinced that changing priorities and shifting towards the legal model would be beneficial and feasible overall? Some scholars in this area, notably Thomas Huddle, are chary of demanding socially focused actions of doctors.

Authors’ response Thank you for pointing this out, we have removed the section: 

“Alongside Earnest et al. (1), we propose that policy-makers could look to the legal sector in emulating a pro-bono target representing principles of medical professionalism.”

In this section we were attempting to expound upon a previous idea regarding pro-bono work mentioned in lines 1118 – 1120 (that professional organisations could require doctors to perform some service work with marginalised groups to maintain certification) by offering a practical means of implementing this idea. However, we agree that the implication of moving towards a legal ‘pro-bono’ model is troublesome, and that this idea in itself was poorly explained. By removing this section, we believe that the original paragraph is improved and simplified. 

We also hope this reconciles the second point of comment 9, as Huddle is positive of community service and pro-bono work as a traditional and historical commitment of the medical profession that does not require a problematic political commitment. He discusses this in, 

Huddle T. Perspective: Medical Professionalism and Medical Education Should Not Involve Commitments to Political Advocacy. Academic Medicine. 2011;86(3):378–83. doi: 10.1097/ACM.0b013e3182086efe. and,

Huddle T. The Limits of Social Justice as an Aspect of Medical Professionalism. Journal of Medicine and Philosophy. 2013;38:369–87. doi: 10.1093/jmp/jht024.

We agree with the reviewer that Huddle’s arguments are highly relevant to our discussion, especially considering how many of our respondents engaged in political advocacy actions (antithetical to Huddle’s position). We have introduced a short segment considering the role of medical advocates at the end of the discussion (lines 1183 – 1196) and make mention of this point there. 

Conclusion: I enjoyed reading this work and consider its clear and thorough description of practitioners’ perspectives on ethical dimensions of work in immigration detention to be illuminating and important.

REVIEWER 2: 

General: The study is of significant importance as it is an area of limited research, particularly the views of medical professionals. The findings have the potential to provide relevant information to the various stakeholders.

R2 comment 1 1. The introduction and background provides adequate information to the reader. The selected literature was appropriate and builds up to the study rationale. Part of the rationale included the importance of the “role of medical advocates” line 183. Perhaps this should be included in the discussion. A paragraph will be adequate as the study findings does have the relevant information to discuss the role of medical advocates.

Authors’ response We agree with the reviewer that this was an important part of the study and should be given more consideration in the discussion. 

A few sentences were also added to ‘Theme 4, Part 2’ of the results section ‘Defining medical advocacy and its scope in systems of abuse’ to clarify respondents’ perceptions around their role as medical advocates, and the role of medical advocacy more generally. Please refer to lines 956 – 964.

As suggested, we have also included a paragraph in the discussion regarding the role of medical advocates in systems that diminish the health rights of patients. We believe this ties the discussion more clearly to the aims as the reviewer proposed, and also works to situate this research in the broader debate on medical advocacy. Please refer to lines 1183 – 1196. 

“The scope and practice of medical advocacy remains the cause of debate among the medical community [5]. While many argue for the extension of medical advocacy to the public sphere [6, 13, 20, 21], others hold that politics has no place in medicine, and that engagement in social justice will necessarily displace clinical duties [22, 23]. This debate is reframed in the setting of Australian immigration detention, with respondents describing how the most fundamental professional responsibilities can become politicised, and how medical advocacy can become essential for the delivery of standard clinical care. These findings are not novel and have been reported in observational literature from clinicians working in IDCs across the world [24]. In systems that diminish patients’ health rights, medical advocates play a crucial role in documenting human rights abuses [12, 24-27] and countering government claims of care equivalence [28, 29]: actions that lie far beyond the scope of traditional clinical medicine. Results of respondents’ delineation of medical advocacy in the political realm are thus illuminating and could provide guidance to clinicians attempting to define or engage in medical advocacy worldwide.”

R2 comment 2 2. Line 191 “shape medical advocates” I understand what you mean, but maybe this should be expanded. Shape them in what way? Also, the authors state “factors that shape” – be specific, for example motivational factors. It will link directly to your findings.

Authors’ response Thank you for this comment, this was unclear in the original manuscript. We have clarified as suggested and line 290 now reads: “Furthermore, by examining motivating factors for participants’ engagement with medical advocacy,”

R2 comment 3 3. Line 193 “participants’ – suggest change to the medical professionals, as you are only referring to them, not students or the refugees themselves.

Authors’ response This has been clarified. Line 292 now reads: “The need for analysis of medical advocate participants’ understandings in their own words means that an in-depth qualitative approach has been necessary for assessment.”

R2 comment 4 4. The aims need to clarified as the aims in the abstract are somewhat different from the aims in the study. 

4.1. I also suggest that the “Research aims” section be moved after the rationale.

Authors’ response 4. Thank you for pointing this out. The research aims have been updated in the manuscript. 

In response to suggestions made by reviewer 1 the first aim now reads: “1. To document the personal views and experiences of Australian medical professionals advocating for the health of refugee and asylum seekers populations in mandatory immigration detention” (Line 311 – 313) 

We have also added a third and fourth aim: 

“3. To provide insight into factors that motivate clinician engagement and perseverance with medical advocacy, and, 

4. To examine the role of medical advocates in systems that diminish patients’ health rights.” (Line 316 – 319). 

We now believe that the research aims appropriately reflect the direction of the manuscript (particularly the “Results” and “Discussion”) and encompass those outlined in the abstract. 

4.1 The “Research aims” section has now been moved after the “Rationale” section of the paper, which improves manuscript structure and clarity. Thank you for this suggestion. 

R2 comment 5 5. The methodology section requires major revision. There are problematic assumptions regarding the methods and rigour. Firstly, the COREQ guidelines were used, which may also be debatable. Please refer to:

Hannes, K., Heyvaert, M., Slegers, K., Vandenbrande, S., & Van Nuland, M. (2015). Exploring the Potential for a Consolidated Standard for Reporting Guidelines for Qualitative Research: An Argument Delphi Approach. International Journal of Qualitative Methods. https://doi.org/10.1177/1609406915611528

5.1. Nevertheless, not all information is reported as per COREQ checklist within the manuscript. For e.g. the methodological orientation and theory - What methodological orientation was stated to underpin the study? e.g. grounded theory, discourse analysis, ethnography, phenomenology, content analysis. This not presented in the data. 

5.2. Furthermore, this orientation should be linked to the study findings in the discussion. (theory, methods, alignment)

Authors’ response 5. We note concerns regarding the COREQ guidelines, these are used in keeping with the recommended guidelines for PLOS One. 

5.1 Thank you for this comment, and apologies for the oversight. We have made some edits and are attaching a sheet that itemises where each item on the COREQ guidelines is address in the paper (see for example pages 10 – 14). 

5.2 The conceptual framework for this study is grounded in empirical bioethics and we attempted to enact internal coherence through reflexivity. In line with this we have made clarifications in the ‘Methodology’ section (refer pages 10 - 14) and ‘Results’ section (refer lines 463 - 472). We have also made reference to this in the discussion to improve clarity:

“In line with the aims of empirical bioethics to ‘bring about ethically defensible changes to practice’ [58], we suggest that each of these three factors should be addressed in future endeavours to promote advocacy within the profession.” (Line 989 – 990). 

R2 comment 6 6. The data analysis, Line 269 “emergent themes” and the Line 275 “Braun and Clark” is controversial. Please read: Braun, V., & Clarke, V. (2019). Reflecting on reflexive thematic analysis. Qualitative Research in Sport, Exercise and Health, 11(4), 589–597. doi:10.1080/2159676x.2019.1628806 

This shows a shift in their (Braun and Clark) thinking from 2006, which poses major flaws in the analysis.

6.1. It is also recommended that the following paper be read: Smith, B., & McGannon, K. R. (2017). Developing rigor in qualitative research: problems and opportunities within sport and exercise psychology. International Review of Sport and Exercise Psychology, 11(1), 101–121. doi:10.1080/1750984x.2017.1317357

The paper needs to demonstrate rigour, hence demonstrating quality, qualitative data analysis.

Authors’ response Thanks for this comment. We recognise that using the term ‘emergent’ is epistemologically at odds with reflexive thematic analysis, and have changed wording at line 489 to better reflect our epistemic position:

“and new codes were generated with successive transcripts”

and at 1218 – 1219:

“The negative consequences of advocacy on personal wellbeing was an unanticipated theme that we identified.”

Please also see new text under ‘Methodology’:

“The conceptual framework for this study is grounded in empirical bioethics, which is an interdisciplinary approach that integrates empirical social science methods with ethical analysis to draw normative conclusions [58]. Data was collected through semi-structured interviews; a foundational and widely used method in qualitative research [59] which facilitates rich insights into subjective perspectives of experience and enables exploration and critical reflection on ethically complex topics [60]. Our analytical approach draws on reflexive thematic analysis of Braun and Clark, which recognises that qualitative analysis is an interpretive process informed by the research team’s assumptions, values and commitments [60-62]. 

Our epistemological perspective is constructivist [63]. Study rigour is demonstrated by the selection of sampling, data collection and analysis methods that were best suited to answering our research questions [64].” (Lines 323 – 334)

R2 comment 7 7. The results appeared to be very long, although was easy to read and understand. 

7.1 The relevance of selected quotes is questionable. There are too many quotations. It is recommended that the results section be condensed. 

7.2 Furthermore, the sub-headings per theme, are these sub-themes? Please clarify.

7.3 As mentioned earlier, the discussion should link to the theoretical framework. Too many subheadings should be avoided in this section. Overall, the structure of the paper needs to be revised.

Authors’ response 7 We appreciate the encouraging comments regarding the readability of the results section. We have attempted to somewhat condense this section by removing some of the less relevant quotes, as below. 

7.1 We appreciate the reviewer’s comments and have removed some of the less relevant quotes, including the poetry quotation at the start of the results section. 

When originally deciding what should be included in the results section, the authors sometimes weighed the direct relevance of quotes against the need for participants to be able to express themselves in their own words, something that we believe to be essential in achieving our third aim: “To document the personal views and experiences of those participating as advocates.” In some cases, eg. Line 621, we felt the need for participant expression outweighed the strict relevance of the quotes to the results. 

7.2 This has been flagged and clarified in lines 463 – 472.

 “The divisions into two parts of theme 3 and 4 reflect important delineations identified in participant’ responses that added to the depth of existing themes, rather than creating new ones.” (Line 469). 

Please refer to R1 comment 7. 

7.3 We have removed subheadings from the ‘Discussion’. We have also reworked this section to increase readability and cohesiveness and included a paragraph discussing the role of medical advocates in systems of abuse and situating this research in the broader debate on medical advocacy (please refer to R2 comment 1). 

R2 comment 8 8. The place/country of study of the health professionals should be taken into account as this may have influenced their responses to questions. It is also relevant since the results are to influence the education of the medical professionals training in Australia. This could be a limitation of the study.

Authors’ response We agree with the reviewer’s comment that the medical training of participants could influence both understandings of medical professionalism and responses to interview questions. We have chosen not to highlight the location of training in this study as we believe that this would add to the complexity of the presentation of results and may prove to be identifying in some cases. 

Only 7 of the 32 participants were trained internationally: 3 in Britain, 2 in Ireland, 1 in South Africa, and 1 in Baghdad. The vast majority (25) trained in Australia, mostly in NSW and Victoria. Of note, of the 7 participants who were trained internationally, 5 are professors or senior lecturers at Australian universities and have been involved in medical teaching in Australia for a number of years, which may also have influenced responses to questions. 

Importantly, no major differences or trends were noted in participant responses comparing these two groups, which may be due largely to the similarity of medical training programs worldwide, particularly those between Australia and the UK. 

There is also the concern that noting the place of study (especially in some cases) would prove to be identifying. Many of the respondents are high-profile individuals and others may be recognisable to those who work in the refugee advocacy space or in particular areas of specialist medicine.

The authors would like to suggest that a study comparing the attitudes of physicians (toward professional roles, social justice and medical advocacy) from a range of countries and training programs could be an interesting direction for future research, and would like to acknowledge that we believe it to be outside of the scope of our study. 

R2 comment 9 Technical considerations: 

9. Asylum seekers vs asylum-seeker

9.1. Well being vs well-being

9.2. P02 vs P02 (italics)

9.3. The indentation of paragraphs-looks “messy”

9.4. Please check for consistency.

9.5. Line 583 “they’d” – this should be in full

Authors’ response Technical considerations have been addressed appropriately throughout the manuscript. 

9 “asylum seeker” in the unhyphenated form has been standardised throughout the manuscript. 

9.1 “wellbeing” as a single word has been standardised throughout the manuscript. 

9.2 P02 has been italicised in line 956 in keeping with the rest of the manuscript. 

9.3 We have removed paragraph indentation throughout and note that paper will likely be type-set in journal style.

9.4 Some small corrections / inconsistencies have been remedied. 

9.5 We agree, “they’d” has been changed to “they would” (line 841).

---

## [Decision Letter · Decision Letter 1]

4 Aug 2020

Medical advocacy in the face of Australian immigration practices: a study of medical professionals defending the health rights of detained refugees and asylum seekers

PONE-D-19-29972R1

Dear Dr. Stoddart,

We’re pleased to inform you that your manuscript has been judged scientifically suitable for publication and will be formally accepted for publication once it meets all outstanding technical requirements.

Kind regards,

Joseph Telfair, DrPH, MSW, MPH

Academic Editor

PLOS ONE

Additional Editor Comments (optional):

Reviewers' comments:

Reviewer's Responses to Questions

**Comments to the Author**

1. If the authors have adequately addressed your comments raised in a previous round of review and you feel that this manuscript is now acceptable for publication, you may indicate that here to bypass the “Comments to the Author” section, enter your conflict of interest statement in the “Confidential to Editor” section, and submit your "Accept" recommendation.

Reviewer #1: (No Response)

Reviewer #2: All comments have been addressed

2. Is the manuscript technically sound, and do the data support the conclusions?

Reviewer #1: Yes

Reviewer #2: Yes

3. Has the statistical analysis been performed appropriately and rigorously? 

Reviewer #1: N/A

Reviewer #2: Yes

4. Have the authors made all data underlying the findings in their manuscript fully available?

Reviewer #1: Yes

Reviewer #2: Yes

5. Is the manuscript presented in an intelligible fashion and written in standard English?

Reviewer #1: Yes

Reviewer #2: Yes

6. Review Comments to the Author

Reviewer #1: Thank you for your responses to my comments on the original manuscript. I feel that they have responded thoroughly to the matters I raised. I have 2 comments on the revision:

In material added since the first draft, lines 152-159, there is a lack of clarity about which of the listed deficiencies relate to asylum seeker care generally (i.e., care by aspects of the system that are not directly related to the maintenance and restoration of health, for example, accommodation, security) and which are directly related to how health care services are provided. This should be clarified before publication. I would also recommend justifying in greater detail the claim that government policy actively opposes meeting standards of (health?) care.

Notwithstanding the significant response to my earlier comment, I remain concerned that using non-doctors’ data has not been adequately justified and I lean towards the view that the paper would have been stronger had these data been excluded. At this stage, I would recommend further clarification on whether the non-doctors were speaking on the subject of doctors’ advocacy or health care professional advocacy generally. It would also help with clarity, I think, if the term ‘health care professional’ were to replace ‘medical professionals’ at line 322 and wherever the latter is used to mean ‘doctors and other health professionals’

Reviewer #2: The authors have addressed all my comments and edits. I am very satisfied with all revisions. The paper is now easy to read and is definitely an important piece of research that contributes to its field. Well done to the research team and or authors.

7. PLOS authors have the option to publish the peer review history of their article (what does this mean?). If published, this will include your full peer review and any attached files.

Reviewer #1: No

Reviewer #2: **Yes: **Prof. Rowena Naidoo

---

## [Editor Report · Acceptance letter]

10 Aug 2020

PONE-D-19-29972R1 

Medical advocacy in the face of Australian immigration practices: a study of medical professionals defending the health rights of detained refugees and asylum seekers 

Dear Dr. Stoddart:

I'm pleased to inform you that your manuscript has been deemed suitable for publication in PLOS ONE. Congratulations! Your manuscript is now with our production department. 

Kind regards, 

on behalf of

Dr. Joseph Telfair 

Academic Editor

PLOS ONE